# Limitations of the Empirical Fisher Approximation for Natural Gradient Descent

**Frederik Kunstner**[1,2,3]
kunstner@cs.ubc.ca

**Lukas Balles**[2,3]
lballes@tue.mpg.de

**Philipp Hennig**[2,3]
ph@tue.mpg.de

École Polytechnique Fédérale de Lausanne (EPFL), Switzerland[1]
University of Tübingen, Germany[2]
Max Planck Institute for Intelligent Systems, Tübingen, Germany[3]

## Abstract

Natural gradient descent, which preconditions a gradient descent update with the Fisher information matrix of the underlying statistical model, is a way to capture partial second-order information. Several highly visible works have advocated an approximation known as the empirical Fisher, drawing connections between approximate second-order methods and heuristics like Adam. We dispute this argument by showing that the empirical Fisher—unlike the Fisher—does not generally capture second-order information. We further argue that the conditions under which the empirical Fisher approaches the Fisher (and the Hessian) are unlikely to be met in practice, and that, even on simple optimization problems, the pathologies of the empirical Fisher can have undesirable effects.

## 1   Introduction

Consider a supervised machine learning problem of predicting outputs $y \in \mathbb{Y}$ from inputs $x \in \mathbb{X}$. We assume a probabilistic model for the conditional distribution of the form $p_\theta(y|x) = p(y|f(x,\theta))$, where $p(y|\cdot)$ is an exponential family with natural parameters in $\mathbb{F}$ and $f \colon \mathbb{X} \times \mathbb{R}^D \to \mathbb{F}$ is a prediction function parameterized by $\theta \in \mathbb{R}^D$. Given $N$ iid training samples $(x_n, y_n)_{n=1}^N$, we want to minimize

$$\mathcal{L}(\theta) := - \textstyle\sum_n \log p_\theta(y_n|x_n) = - \textstyle\sum_n \log p(y_n|f(x_n, \theta)). \tag{1}$$

This framework covers common scenarios such as least-squares regression ($\mathbb{Y} = \mathbb{F} = \mathbb{R}$ and $p(y|f) = \mathcal{N}(y; f, \sigma^2)$ with fixed $\sigma^2$) or $C$-class classification with cross-entropy loss ($\mathbb{Y} = \{1, \ldots, C\}$, $\mathbb{F} = \mathbb{R}^C$ and $p(y = c|f) = \exp(f_c)/\sum_i \exp(f_i)$) with an arbitrary prediction function $f$. Eq. (1) can be minimized by gradient descent, which updates $\theta_{t+1} = \theta_t - \gamma_t \nabla \mathcal{L}(\theta_t)$ with step size $\gamma_t \in \mathbb{R}$. This update can be *preconditioned* with a matrix $B_t$ that incorporates additional information, such as local curvature, $\theta_{t+1} = \theta_t - \gamma_t B_t^{-1} \nabla \mathcal{L}(\theta_t)$. Choosing $B_t$ to be the Hessian yields Newton's method, but its computation is often burdensome and might not be desirable for non-convex problems. A prominent variant in machine learning is *natural gradient descent* [NGD; Amari, 1998]. It adapts to the *information geometry* of the problem by measuring the distance between parameters with the Kullback–Leibler divergence between the resulting distributions rather than their Euclidean distance, using the Fisher information matrix (or simply "Fisher") of the model as a preconditioner,

$$\mathrm{F}(\theta) := \textstyle\sum_n \mathbb{E}_{p_\theta(y|x_n)} \left[ \nabla_\theta \log p_\theta(y|x_n) \, \nabla_\theta \log p_\theta(y|x_n)^\top \right]. \tag{2}$$

While this motivation is conceptually distinct from approximating the Hessian, the Fisher coincides with a generalized Gauss-Newton [Schraudolph, 2002] approximation of the Hessian for the problems presented here. This gives NGD theoretical grounding as an approximate second-order method.

A number of recent works in machine learning have relied on a certain approximation of the Fisher, which is often called the *empirical Fisher (EF)* and is defined as

$$\widetilde{\mathrm{F}}(\theta) := \textstyle\sum_n \nabla_\theta \log p_\theta(y_n|x_n) \, \nabla_\theta \log p_\theta(y_n|x_n)^\top. \tag{3}$$

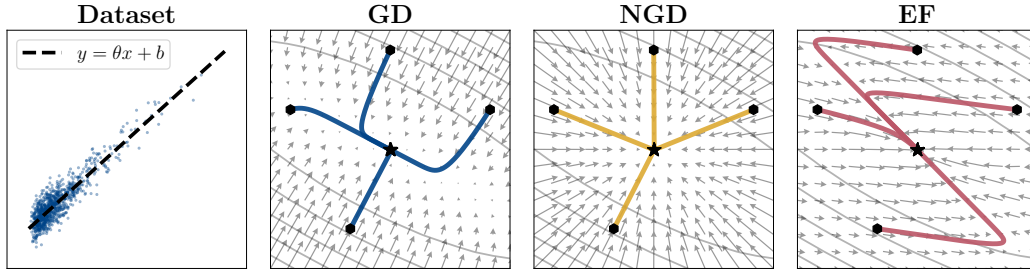

Figure 1: Fisher vs. empirical Fisher as preconditioners for linear least-squares regression on the data shown in the left-most panel. The second plot shows the gradient vector field of the (quadratic) loss function and sample trajectories for gradient descent. The remaining plots depict the vector fields of the natural gradient and the "EF-preconditioned" gradient, respectively. NGD successfully adapts to the curvature whereas preconditioning with the empirical Fisher results in a distorted gradient field.

At first glance, this approximation is merely replacing the expectation over $y$ in Eq. (2) with a sample $y_n$. However, $y_n$ is a training label and *not* a sample from the model's predictive distribution $p_\theta(y|x_n)$. Therefore, and contrary to what its name suggests, the empirical Fisher is *not* an empirical (i.e. Monte Carlo) estimate of the Fisher. Due to the unclear relationship between the model distribution and the data distribution, the theoretical grounding of the empirical Fisher approximation is dubious.

Adding to the confusion, the term "empirical Fisher" is used by different communities to refer to different quantities. Authors closer to statistics tend to use "empirical Fisher" for Eq. (2), while many works in machine learning, some listed in Section 2, use "empirical Fisher" for Eq. (3). While the statistical terminology is more accurate, we adopt the term "Fisher" for Eq. (2) and "empirical Fisher" for Eq. (3), which is the subject of this work, to be accessible to readers more familiar with this convention. We elaborate on the different uses of the terminology in Section 3.1.

The main purpose of this work is to provide **a detailed critical discussion of the empirical Fisher approximation.** While the discrepancy between the empirical Fisher and the Fisher has been mentioned in the literature before [Pascanu and Bengio, 2014, Martens, 2014], we see the need for a detailed elaboration of the subtleties of this important issue. The intricacies of the relationship between the empirical Fisher and the Fisher remain opaque from the current literature. Not all authors using the EF seem to be fully aware of the heuristic nature of this approximation and overlook its shortcomings, which can be seen clearly even on simple linear regression problems, see Fig. 1. Natural gradients adapt to the curvature of the function using the Fisher while the empirical Fisher distorts the gradient field in a way that lead to worse updates than gradient descent.

The empirical Fisher approximation is so ubiquitous that it is sometimes just called the Fisher [e.g., Chaudhari et al., 2017, Wen et al., 2019]. Possibly as a result of this, there are examples of algorithms involving the Fisher, such as Elastic Weight Consolidation [Kirkpatrick et al., 2017] and KFAC [Martens and Grosse, 2015], which have been re-implemented by third parties using the empirical Fisher. Interestingly, there is also at least one example of an algorithm that was originally developed using the empirical Fisher and later found to work better with the Fisher [Wierstra et al., 2008, Sun et al., 2009]. As the empirical Fisher is now used beyond optimization, for example as an approximation of the Hessian in empirical works studying properties of neural networks [Chaudhari et al., 2017, Jastrzębski et al., 2018], the pathologies of the EF approximation may lead the community to erroneous conclusions—an arguably more worrisome outcome than a suboptimal preconditioner.

The poor theoretical grounding stands in stark contrast to the practical success that empirical Fisher-based methods have seen. This paper is in no way meant to negate these practical advances but rather points out that the existing justifications for the approximation are insufficient and do not stand the test of simple examples. This indicates that there are effects at play that currently elude our understanding, which is not only unsatisfying, but might also prevent advancement of these methods. We hope that this paper helps spark interest in understanding these effects; our final section explores a possible direction.

## 1.1 Overview and contributions

We first provide a short but complete overview of natural gradient and the closely related generalized Gauss-Newton method. Our main contribution is a critical discussion of the *specific arguments* used to advocate the empirical Fisher approximation. A principal conclusion is that, while the empirical Fisher follows the formal definition of a generalized Gauss-Newton matrix, it is not guaranteed to capture any useful second-order information. We propose a clarifying amendment to the definition of a generalized Gauss-Newton to ensure that all matrices satisfying it have useful approximation properties. Furthermore, while there are conditions under which the empirical Fisher approaches the true Fisher, we argue that these are unlikely to be met in practice. We illustrate that using the empirical Fisher can lead to highly undesirable effects; Fig. 1 shows a first example.

This raises the question: Why are methods based on the empirical Fisher practically successful? We point to an alternative explanation, as an adaptation to gradient noise in *stochastic* optimization instead of an adaptation to curvature.

## 2 Related work

The generalized Gauss-Newton [Schraudolph, 2002] and natural gradient descent [Amari, 1998] methods have inspired a line of work on approximate second-order optimization [Martens, 2010, Botev et al., 2017, Park et al., 2000, Pascanu and Bengio, 2014, Ollivier, 2015]. A successful example in modern deep learning is the KFAC algorithm [Martens and Grosse, 2015], which uses a computationally efficient structural approximation to the Fisher.

Numerous papers have relied on the empirical Fisher approximation for preconditioning and other purposes. Our critical discussion is in no way intended as an invalidation of these works. All of them provide important insights and the use of the empirical Fisher is usually not essential to the main contribution. However, there is a certain degree of vagueness regarding the relationship between the Fisher, the EF, Gauss-Newton matrices and the Hessian. Oftentimes, only limited attention is devoted to possible implications of the empirical Fisher approximation.

The most prominent example of preconditioning with the EF is Adam, which uses a moving average of squared gradients as "an approximation to the diagonal of the Fisher information matrix" [Kingma and Ba, 2015]. The EF has been used in the context of variational inference by various authors [Graves, 2011, Zhang et al., 2018, Salas et al., 2018, Khan et al., 2018, Mishkin et al., 2018], some of which have drawn further connections between NGD and Adam. There are also several works building upon KFAC which substitute the EF for the Fisher [George et al., 2018, Osawa et al., 2019].

The empirical Fisher has also been used as an approximation of the Hessian for purposes other than preconditioning. Chaudhari et al. [2017] use it to investigate curvature properties of deep learning training objectives. It has also been employed to explain certain characteristics of SGD [Zhu et al., 2019, Jastrzębski et al., 2018] or as a diagnostic tool during training [Liao et al., 2020].

Le Roux et al. [2007] and Le Roux and Fitzgibbon [2010] have considered the empirical Fisher in its interpretation as the (non-central) covariance matrix of stochastic gradients. While they refer to their method as "Online Natural Gradient", their goal is explicitly to adapt the update to the *stochasticity* of the gradient estimate, *not to curvature*. We will return to this perspective in Section 5.

Before moving on, we want to re-emphasize that other authors have previously raised concerns about the empirical Fisher approximation [e.g., Pascanu and Bengio, 2014, Martens, 2014]. This paper is meant as a detailed elaboration of this known but subtle issue, with novel results and insights. Concurrent to our work, Thomas et al. [2019] investigated similar issues in the context of estimating the generalization gap using information criteria.

## 3 Generalized Gauss-Newton and natural gradient descent

This section briefly introduces natural gradient descent, adresses the difference in terminology for the quantities of interest across fields, introduces the generalized Gauss-Newton (GGN) and reviews the connections between the Fisher, the GGN, and the Hessian.

| Quantity | | Terminology in statistics | and machine learning |
|---|---|---|---|
| $F_{\prod_n p_\theta(x,y)}$ | Eq. (5) | Fisher | |
| $F_{\prod_n p_\theta(y\|x_n)}$ | Eq. (6) | empirical Fisher | Fisher |
| $\tilde{F}$ | Eq. (7) | | empirical Fisher |

Table 1: Common terminology for the Fisher information and related matrices by authors closely aligned with statistics, such as Amari [1998], Park et al. [2000], and Karakida et al. [2019], or machine learning, such as Martens [2010], Schaul et al. [2013], and Pascanu and Bengio [2014].

## 3.1 Natural gradient descent

Gradient descent follows the direction of "steepest descent", the negative gradient. But the definition of *steepest* depends on a notion of distance and the gradient is defined with respect to the Euclidean distance. The natural gradient is a concept from information geometry [Amari, 1998] and applies when the gradient is taken w.r.t. the parameters $\theta$ of a probability distribution $p_\theta$. Instead of measuring the distance between parameters $\theta$ and $\theta'$ with the Euclidean distance, we use the Kullback–Leibler (KL) divergence between the distributions $p_\theta$ and $p_{\theta'}$. The resulting steepest descent direction is the negative gradient preconditioned with the Hessian of the KL divergence, which is exactly the *Fisher information matrix* of $p_\theta$,

$$F(\theta) := \mathbb{E}_{p_\theta(z)}\left[\nabla_\theta \log p_\theta(z) \nabla_\theta \log p_\theta(z)^T\right] = \mathbb{E}_{p_\theta(z)}\left[-\nabla_\theta^2 \log_\theta p(z)\right]. \tag{4}$$

The second equality may seem counterintuitive; the difference between the outer product of gradients and the Hessian cancels out in expectation with respect to the model distribution at $\theta$, see Appendix A. This equivalence highlights the relationship of the Fisher to the Hessian.

## 3.2 Difference in terminology across fields

In our setting, we only model the conditional distribution $p_\theta(y|x)$ of the joint distribution $p_\theta(x,y) = p(x)p_\theta(y|x)$. The Fisher information of $\theta$ for $N$ samples from the joint distribution $p_\theta(x,y)$ is

$$F_{\prod_n p_\theta(x,y)}(\theta) = N \, \mathbb{E}_{x,y \sim p(x)p_\theta(y|x)}\left[\nabla_\theta \log p_\theta(y|x) \nabla_\theta \log p_\theta(y|x)^T\right], \tag{5}$$

This is what statisticians would call the "Fisher information" of the model $p_\theta(x,y)$. However, we typically do not know the distribution over inputs $p(x)$, so we use the empirical distribution over $x$ instead and compute the Fisher information of the conditional distribution $\prod_n p_\theta(y|x_n)$;

$$F_{\prod_n p_\theta(y|x_n)}(\theta) = \sum_n \mathbb{E}_{y \sim p_\theta(y|x_n)}\left[\nabla_\theta \log p_\theta(y|x_n) \nabla_\theta \log p_\theta(y|x_n)^T\right]. \tag{6}$$

This is Eq. (2), which we call the "Fisher". This is the terminology used by work on the application of natural gradient methods in machine learning, such as Martens [2014] and Pascanu and Bengio [2014], as it is the Fisher information for the distribution we are optimizing, $\prod_n p_\theta(y|x_n)$. Work closer to the statistics literature, following the seminal paper of Amari [1998], such as Park et al. [2000] and Karakida et al. [2019], call this quantity the "empirical Fisher" due to the usage of the empirical samples for the inputs. In constrast, we call Eq. (3) the "empirical Fisher", restated here,

$$\tilde{F}(\theta) = \sum_n \nabla_\theta \log p_\theta(y_n|x_n) \nabla_\theta \log p_\theta(y_n|x_n)^T, \tag{7}$$

where "empirical" describes the use of samples for both the inputs and the outputs. This expression, however, does not have a direct interpretation as a Fisher information as it does not sample the output according to the distribution defined by the model. Neither is it a Monte-Carlo approximation of Eq. (6), as the samples $y_n$ do not come from $p_\theta(y|x_n)$ but from the data distribution $p(y|x_n)$. How close the empirical Fisher (Eq. 7) is to the Fisher (Eq. 6) depends on how close the model $p_\theta(y|x_n)$ is to the true data-generating distribution $p(y|x_n)$.

## 3.3 Generalized Gauss-Newton

One line of argument justifying the use of the empirical Fisher approximation uses the connection between the Hessian and the Fisher through the generalized Gauss-Newton (GGN) matrix [Schraudolph, 2002]. We give here a condensed overview of the definition and properties of the GGN.

The original Gauss-Newton algorithm is an approximation to Newton's method for nonlinear least squares problems, $\mathcal{L}(\theta) = \frac{1}{2}\sum_n (f(x_n, \theta) - y_n)^2$. By the chain rule, the Hessian can be written as

$$\nabla^2 \mathcal{L}(\theta) = \underbrace{\sum_n \nabla_\theta f(x_n, \theta)\nabla_\theta f(x_n, \theta)^\top}_{:=G(\theta)} + \underbrace{\sum_n r_n \nabla^2_\theta f(x_n, \theta)}_{:=R(\theta)}, \qquad (8)$$

where $r_n = f(x_n, \theta) - y_n$ are the residuals. The first part, $G(\theta)$, is the Gauss-Newton matrix. For small residuals, $R(\theta)$ will be small and $G(\theta)$ will approximate the Hessian. In particular, when the model perfectly fits the data, the Gauss-Newton is equal to the Hessian.

Schraudolph [2002] generalized this idea to objectives of the form $\mathcal{L}(\theta) = \sum_n a_n(b_n(\theta))$, with $b_n \colon \mathbb{R}^D \to \mathbb{R}^M$ and $a_n \colon \mathbb{R}^M \to \mathbb{R}$, for which the Hessian can be written as[1]

$$\nabla^2 \mathcal{L}(\theta) = \sum_n (\mathrm{J}_\theta b_n(\theta))^\top \ \nabla^2_b a_n(b_n(\theta)) \ (\mathrm{J}_\theta b_n(\theta)) + \sum_{n,m} [\nabla_b a_n(b_n(\theta))]_m \nabla^2_\theta b_n^{(m)}(\theta). \quad (9)$$

The generalized Gauss-Newton matrix (GGN) is defined as the part of the Hessian that ignores the second-order information of $b_n$,

$$G(\theta) := \sum_n [\mathrm{J}_\theta b_n(\theta)]^\top \ \nabla^2_b a_n(b_n(\theta)) \ [\mathrm{J}_\theta b_n(\theta)]. \qquad (10)$$

If $a_n$ is convex, as is customary, the GGN is positive (semi-)definite even if the Hessian itself is not, making it a popular curvature matrix in non-convex problems such as neural network training. The GGN is ambiguous as it crucially depends on the "split" given by $a_n$ and $b_n$. As an example, consider the two following possible splits for the least-squares problem from above:

$$a_n(b) = \tfrac{1}{2}(b - y_n)^2, \ b_n(\theta) = f(x_n, \theta), \quad \text{or} \quad a_n(b) = \tfrac{1}{2}(f(x_n, b) - y_n)^2, \ b_n(\theta) = \theta. \quad (11)$$

The first recovers the classical Gauss-Newton, while in the second case, the GGN equals the Hessian. While this is an extreme example, the split will be important for our discussion.

### 3.4 Connections between the Fisher, the GGN and the Hessian

While NGD is not explicitly motivated as an approximate second-order method, the following result, noted by several authors,[2] shows that the Fisher captures partial curvature information about the problem defined in Eq. (1).

**Proposition 1** (Martens [2014], §9.2). *If $p(y|f)$ is an exponential family distribution with natural parameters $f$, then the Fisher information matrix coincides with the GGN of Eq. (1) using the split*

$$a_n(b) = -\log p(y_n|b), \qquad\qquad b_n(\theta) = f(x_n, \theta), \qquad (12)$$

*and reads* $\mathrm{F}(\theta) = G(\theta) = -\sum_n [\mathrm{J}_\theta f(x_n, \theta)]^\top \ \nabla^2_f \log p(y_n|f(x_n, \theta)) \ [\mathrm{J}_\theta f(x_n, \theta)]$.

For completeness, a proof can be found in Appendix A. The key insight is that $\nabla^2_f \log p(y|f)$ does not depend on $y$ for exponential families. One can see Eq. (12) as the "canonical" split, since it matches the classical Gauss-Newton for the probabilistic interpretation of least-squares. From now on, when referencing "the GGN" without further specification, we mean this particular split.

The GGN, and under the assumptions of Proposition 1 also the Fisher, are well-justified approximations of the Hessian and we can bound their approximation error in terms of the (generalized) residuals, mirroring the motivation behind the classical Gauss-Newton (Proof in Appendix C.2).

**Proposition 2.** *Let $\mathcal{L}(\theta)$ be defined as in Eq. (1) with $\mathbb{F} = \mathbb{R}^M$. Denote by $f_n^{(m)}$ the $m$-th component of $f(x_n, \cdot) \colon \mathbb{R}^D \to \mathbb{R}^M$ and assume each $f_n^{(m)}$ is $\beta$-smooth. Let $G(\theta)$ be the GGN (Eq. 10). Then,*

$$\|\nabla^2 \mathcal{L}(\theta) - G(\theta)\|_2^2 \le r(\theta)\beta, \qquad (13)$$

*where $r(\theta) = \sum_{n=1}^N \|\nabla_f \log p(y_n|f(x_n, \theta))\|_1$ and $\|\cdot\|_2$ denotes the spectral norm.*

The approximation improves as the residuals in $r(\theta)$ diminish, and is exact if the data is perfectly fit.

# 4 Critical discussion of the empirical Fisher

Two arguments have been put forward to advocate the empirical Fisher approximation. Firstly, it has been argued that it follows the definition of a generalized Gauss-Newton matrix, making it an approximate curvature matrix in its own right. We examine this relation in §4.1 and show that, while technically correct, it does not entail the approximation guarantee usually associated with the GGN.

Secondly, a popular argument is that the empirical Fisher approaches the Fisher at a minimum if the model "is a good fit for the data". We discuss this argument in §4.2 and point out that it requires strong additional assumptions, which are unlikely to be met in practical scenarios. In addition, this argument only applies close to a minimum, which calls into question the usefulness of the empirical Fisher in optimization. We discuss this in §4.3, showing that preconditioning with the empirical Fisher leads to adverse effects on the scaling and the direction of the updates far from an optimum.

We use simple examples to illustrate our arguments. We want to emphasize that, as these are *counter-examples* to arguments found in the existing literature, they are designed to be as simple as possible, and deliberately do not involve intricate state-of-the art models that would complicate analysis. On a related note, while contemporary machine learning often relies on *stochastic* optimization, we restrict our considerations to the deterministic (full-batch) setting to focus on the adaptation to curvature.

## 4.1 The empirical Fisher as a generalized Gauss-Newton matrix

The first justification for the empirical Fisher is that it matches the construction of a generalized Gauss-Newton (Eq. 10) using the split [Bottou et al., 2018]

$$a_n(b) = -\log b, \qquad\qquad b_n(\theta) = p(y_n|f(x_n, \theta)). \qquad (14)$$

Although technically correct,[3] we argue that this split does not provide a reasonable approximation.

For example, consider a least-squares problem which corresponds to the log-likelihood $\log p(y|f) = \log \exp[-\frac{1}{2}(y-f)^2]$. In this case, Eq. (14) splits the identity function, $\log \exp(\cdot)$, and takes into account the curvature from the $\log$ while ignoring that of $\exp$. This questionable split runs counter to the basic motivation behind the classical Gauss-Newton matrix, that small residuals lead to a good approximation to the Hessian: The empirical Fisher

$$\widetilde{F}(\theta) = \sum_n \nabla_\theta \log p_\theta(y_n|x_n) \, \nabla_\theta \log p_\theta(y_n|x_n)^\top = \sum_n r_n^2 \, \nabla_\theta f(x_n, \theta) \, \nabla_\theta f(x_n, \theta)^\top, \qquad (15)$$

approaches zero as the residuals $r_n = f(x_n, \theta) - y_n$ become small. In that same limit, the Fisher $F(\theta) = \sum_n \nabla f(x_n, \theta) \nabla f(x_n, \theta)^\top$ does approach the Hessian, which we recall from Eq. (8) to be given by $\nabla^2 \mathcal{L}(\theta) = F(\theta) + \sum_n r_n \nabla_\theta^2 f(x_n, \theta)$. This argument generally applies for problems where we can fit all training samples such that $\nabla_\theta \log p_\theta(y_n|x_n) = 0$ for all $n$. In such cases, the EF goes to zero while the Fisher (and the corresponding GGN) approaches the Hessian (Prop. 2).

For the generalized Gauss-Newton, the role of the "residual" is played by the gradient $\nabla_b a_n(b)$; compare Equations (8) and (9). To retain the motivation behind the classical Gauss-Newton, the split should be chosen such that this gradient can in principle attain zero, in which case the residual curvature not captured by the GGN in (9) vanishes. The EF split (Eq. 14) does not satisfy this property, as $\nabla_b \log b$ can never go to zero for a probability $b \in [0, 1]$. It might be desirable to amend the definition of a generalized Gauss-Newton to enforce this property (addition in **bold**):

**Definition 1** (Generalized Gauss-Newton). *A split $\mathcal{L}(\theta) = \sum_n a_n(b_n(\theta))$ with convex $a_n$, leads to a generalized Gauss-Newton matrix of $\mathcal{L}$, defined as*

$$G(\theta) = \sum_n G_n(\theta), \qquad\qquad G_n(\theta) := [\mathrm{J}_\theta b_n(\theta)]^\top \ \nabla_b^2 a_n(b_n(\theta)) \ [\mathrm{J}_\theta b_n(\theta)], \qquad (16)$$

***if the split*** $a_n, b_n$ ***is such that there is*** $b_n^* \in \mathrm{Im}(b_n)$ ***such that*** $\nabla_b a_n(b)|_{b=b_n^*} = 0$***.***

Under suitable smoothness conditions, a split satisfying this condition will have a meaningful error bound akin to Proposition 2. To avoid confusion, we want to note that this condition does not assume the existence of $\theta^*$ such that $b_n(\theta^*) = b_n^*$ for all $n$; only that the residual gradient for each data point can, in principle, go to zero.

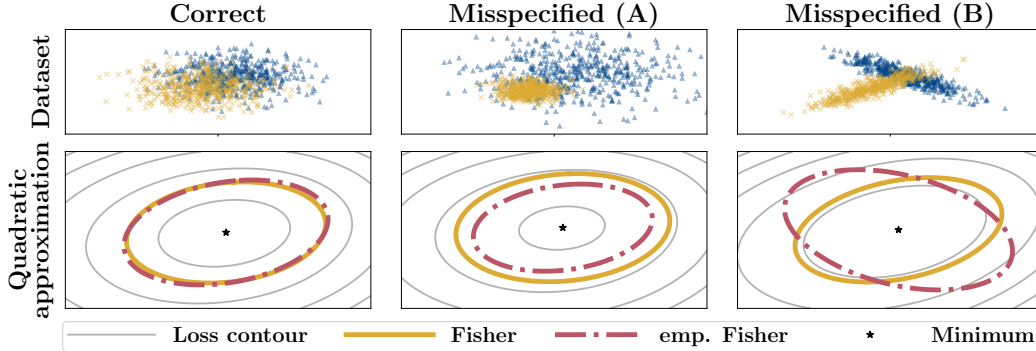

Figure 2: Quadratic approximations of the loss function using the Fisher and the empirical Fisher on a logistic regression problem. Logistic regression implicitly assumes identical class-conditional covariances [Hastie et al., 2009, §4.4.5]. The EF is a good approximation of the Fisher at the minimum if this assumption is fulfilled (left panel), but can be arbitrarily wrong if the assumption is violated, even at the minimum and with large $N$. Note: we achieve classification accuracies of $\geq 85\%$ in the misspecified cases compared to $73\%$ in the well-specified case, which shows that a well-performing model is not necessarily a well-specified one.

## 4.2 The empirical Fisher near a minimum

An often repeated argument is that the empirical Fisher converges to the true Fisher when the model is a good fit for the data [e.g., Jastrzębski et al., 2018, Zhu et al., 2019]. Unfortunately, this is often misunderstood to simply mean "near the minimum". The above statement has to be carefully formalized and requires additional assumptions, which we detail in the following.

Assume that the training data consists of iid samples from some data-generating distribution $p_{\text{true}}(x, y) = p_{\text{true}}(y|x)p_{\text{true}}(x)$. If the model is realizable, i.e., there exists a parameter setting $\theta_{\text{T}}$ such that $p_{\theta_{\text{T}}}(y|x) = p_{\text{true}}(y|x)$, then clearly by a Monte Carlo sampling argument, as the number of data points $N$ goes to infinity, $\widetilde{\text{F}}(\theta_{\text{T}})/N \to \text{F}(\theta_{\text{T}})/N$. Additionally, if the maximum likelihood estimate for $N$ samples $\theta_N^\star$ is consistent in the sense that $p_{\theta_N^\star}(y|x)$ converges to $p_{\text{true}}(y|x)$ as $N \to \infty$,

$$\frac{1}{N}\widetilde{\text{F}}(\theta_N^\star) \overset{N \to \infty}{\longrightarrow} \frac{1}{N}\text{F}(\theta_N^\star). \tag{17}$$

That is, the empirical Fisher converges to the Fisher *at* the minimum as the number of data points grows. (Both approach the Hessian, as can be seen from the second equality in Eq. 4 and detailed in Appendix C.2.) For the EF to be a useful approximation, we thus need (i) a "correctly-specified" model in the sense of the realizability condition, and (ii) enough data to recover the true parameters.

Even under the assumption that $N$ is sufficiently large, the model needs to be able to realize the true data distribution. This requires that the likelihood $p(y|f)$ is well-specified and that the prediction function $f(x, \theta)$ captures all relevant information. This is possible in classical statistical modeling of, say, scientific phenomena where the effect of $x$ on $y$ is modeled based on domain knowledge. But it is unlikely to hold when the model is only approximate, as is most often the case in machine learning. Figure 2 shows examples of model misspecification and the effect on the empirical and true Fisher.

It is possible to satisfy the realizability condition by using a very flexible prediction function $f(x, \theta)$, such as a deep network. However, "enough" data has to be seen relative to the model capacity. The massively overparameterized models typically used in deep learning are able to fit the training data almost perfectly, even when regularized [Zhang et al., 2017]. In such settings, the individual gradients, and thus the EF, will be close to zero at a minimum, whereas the Hessian will generally be nonzero.

## 4.3 Preconditioning with the empirical Fisher far from an optimum

The relationship discussed in §4.2 only holds close to the minimum. Any similarity between $p_\theta(y|x)$ and $p_{\text{true}}(y|x)$ is *very* unlikely when $\theta$ has not been adapted to the data, for example, at the beginning of an optimization procedure. This makes the empirical Fisher a questionable preconditioner.

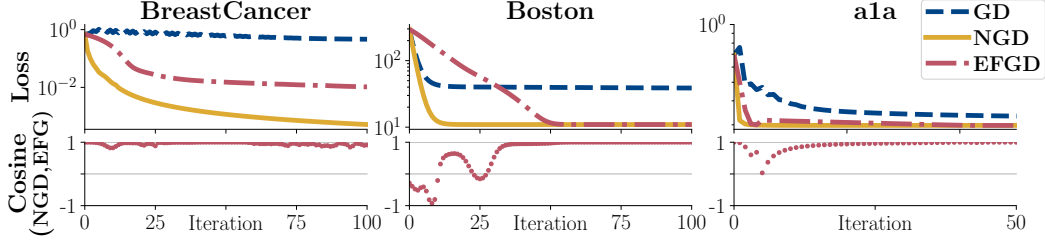

Figure 3: Fisher (NGD) vs. empirical Fisher (EFGD) as preconditioners (with damping) on linear classification (BreastCancer, a1a) and regression (Boston). While the EF *can* be a good approximation for preconditioning on some problems (e.g., a1a), it is not guaranteed to be. The second row shows the cosine similarity between the EF direction and the natural gradient, over the path taken by EFGD, showing that the EF can lead to update directions that are opposite to the natural gradient (see Boston). Even when the direction is correct, the magnitude of the steps can lead to poor performance (see BreastCancer). See Appendix D for details and additional experiments.

In fact, the empirical Fisher can cause severe, adverse distortions of the gradient field far from the optimum, as evident even on the elementary linear regression problem of Fig. 1. As a consequence, EF-preconditioned gradient descent compares unfavorably to NGD even on simple linear regression and classification tasks, as shown in Fig. 3. The cosine similarity plotted in Fig. 3 shows that the empirical Fisher can be arbitrarily far from the Fisher in that the two preconditioned updates point in almost opposite directions.

One particular issue is the scaling of EF-preconditioned updates. As the empirical Fisher is the sum of "squared" gradients (Eq. 3), multiplying the gradient by the inverse of the EF leads to updates of magnitude almost inversely proportional to that of the gradient, at least far from the optimum. This effect has to be counteracted by adapting the step size, which requires manual tuning and makes the selected step size dependent on the starting point; we explore this aspect further in Appendix E.

## 5 Variance adaptation

The previous sections have shown that, interpreted as a *curvature* matrix, the empirical Fisher is a questionable choice at best. Another perspective on the empirical Fisher is that, in contrast to the Fisher, it contains useful information to adapt to the gradient noise in *stochastic* optimization.

In stochastic gradient descent [SGD; Robbins and Monro, 1951], we sample $n \in [N]$ uniformly at random and use a stochastic gradient $g(\theta) = -N \nabla_\theta \log p_\theta(y_n|x_n)$ as an inexpensive but noisy estimate of $\nabla \mathcal{L}(\theta)$. The empirical Fisher, as a sum of outer products of individual gradients, coincides with the non-central second moment of this estimate and can be written as

$$N\widetilde{\mathrm{F}}(\theta) = \Sigma(\theta) + \nabla \mathcal{L}(\theta) \nabla \mathcal{L}(\theta)^\top, \qquad \Sigma(\theta) := \mathbf{cov}[g(\theta)]. \qquad (18)$$

Gradient noise is a major hindrance to SGD and the covariance information encoded in the EF may be used to attenuate its harmful effects, e.g., by scaling back the update in high-noise directions.

A small number of works have explored this idea before. Le Roux et al. [2007] showed that the update direction $\Sigma(\theta)^{-1}g(\theta)$ maximizes the probability of decreasing in function value, while Schaul et al. [2013] proposed a diagonal rescaling based on the signal-to-noise ratio of each coordinate, $D_{ii} := [\nabla \mathcal{L}(\theta)]_i^2 / ([\nabla \mathcal{L}(\theta)]_i^2 + \Sigma(\theta)_{ii})$. Balles and Hennig [2018] identified these factors as *optimal* in that they minimize the expected error $\mathbb{E}\left[\|Dg(\theta) - \nabla \mathcal{L}(\theta)\|_2^2\right]$ for a diagonal matrix $D$.

A straightforward extension of this argument to full matrices yields the variance adaptation matrix

$$M = \left(\Sigma(\theta) + \nabla \mathcal{L}(\theta) \nabla \mathcal{L}(\theta)^\top\right)^{-1} \nabla \mathcal{L}(\theta) \nabla \mathcal{L}(\theta)^\top = (N\widetilde{\mathrm{F}}(\theta))^{-1}\nabla \mathcal{L}(\theta) \nabla \mathcal{L}(\theta)^\top. \qquad (19)$$

In that sense, preconditioning with the empirical Fisher can be understood as an adaptation to gradient noise instead of an adaptation to curvature. The multiplication with $\nabla \mathcal{L}(\theta)\nabla \mathcal{L}(\theta)^\top$ in Eq. (19) will counteract the poor scaling discussed in §4.3.

This perspective on the empirical Fisher is currently not well studied. Of course, there are obvious difficulties ahead: Computing the matrix in Eq. (19) requires the evaluation of all gradients, which

defeats its purpose. It is not obvious how to obtain meaningful estimates of this matrix from, say, a mini-batch of gradients, that would provably attenuate the effects of gradient noise. Nevertheless, we believe that variance adaptation is a possible explanation for the practical success of existing methods using the EF and an interesting avenue for future research. To put it bluntly: it may just be that the name "empirical Fisher" is a fateful historical misnomer, and the quantity should instead just be described as the gradient's non-central second moment.

As a final comment, it is worth pointing out that some methods precondition with the *square-root* of the EF, the prime example being Adam. While this avoids the "inverse gradient" scaling discussed in §4.3, it further widens the conceptual gap between those methods and natural gradient. In fact, such a preconditioning effectively cancels out the gradient magnitude, which has recently been examined more closely as "sign gradient descent" [Balles and Hennig, 2018, Bernstein et al., 2018].

## 6 Conclusions

We offered a critical discussion of the empirical Fisher approximation, summarized as follows:

- While the EF follows the formal definition of a generalized Gauss-Newton matrix, the underlying split does not retain useful second-order information. We proposed a clarifying amendment to the definition of the GGN.

- A clear relationship between the empirical Fisher and the Fisher only exists at a minimum under strong additional assumptions: (i) a correct model and (ii) enough data relative to model capacity. These conditions are unlikely to be met in practice, especially when using overparametrized general function approximators and settling for approximate minima.

- Far from an optimum, EF preconditioning leads to update magnitudes which are inversely proportional to that of the gradient, complicating step size tuning and often leading to poor performance even for linear models.

- As a possible alternative explanation of the practical success of EF preconditioning, and an interesting avenue for future research, we have pointed to the concept of variance adaptation.

The existing arguments do not justify the empirical Fisher as a reasonable approximation to the Fisher or the Hessian. Of course, this does not rule out the existence of certain model classes for which the EF might give reasonable approximations. However, as long as we have not clearly identified and understood these cases, the true Fisher is the "safer" choice as a curvature matrix and should be preferred in virtually all cases.

Contrary to conventional wisdom, the Fisher is not inherently harder to compute than the EF. As shown by Martens and Grosse [2015], an unbiased estimate of the true Fisher can be obtained at the same computational cost as the empirical Fisher by replacing the expectation in Eq. (2) with a single sample $\tilde{y}_n$ from the model's predictive distribution $p_\theta(y|x_n)$. Even exact computation of the Fisher is feasible in many cases. We discuss computational aspects further in Appendix B. The apparent reluctance to compute the Fisher might have more to do with the current lack of convenient implementations in deep learning libraries. We believe that it is misguided—and potentially dangerous—to accept the poor theoretical grounding of the EF approximation purely for implementational convenience.

**Acknowledgements**

We thank Matthias Bauer, Felix Dangel, Filip de Roos, Diego Fioravanti, Jason Hartford, Si Kai Lee, and Frank Schneider for their helpful comments on the manuscript. We thank Emtiyaz Khan, Aaron Mishkin, and Didrik Nielsen for many insightful conversations that lead to this work, and the anonymous reviewers for their constructive feedback.

Lukas Balles kindly acknowledges the support of the International Max Planck Research School for Intelligent Systems (IMPRS-IS). The authors gratefully acknowledge financial support by the European Research Council through ERC StG Action 757275 / PANAMA and the DFG Cluster of Excellence "Machine Learning - New Perspectives for Science", EXC 2064/1, project number 390727645, the German Federal Ministry of Education and Research (BMBF) through the Tübingen AI Center (FKZ: 01IS18039A) and funds from the Ministry of Science, Research and Arts of the State of Baden-Württemberg.

## Footnotes

[1] $\mathrm{J}_\theta b_n(\theta) \in \mathbb{R}^{M \times D}$ is the Jacobian of $b_n$; we use the shortened notation $\nabla^2_b a_n(b_n(\theta)) := \nabla^2_b a_n(b)|_{b=b_n(\theta)}$; $[\cdot]_m$ selects the $m$-th component of a vector; and $b_n^{(m)}$ denotes the $m$-th component function of $b_n$.

[2] Heskes [2000] showed this for regression with squared loss, Pascanu and Bengio [2014] for classification with cross-entropy loss, and Martens [2014] for general exponential families. However, this has been known earlier in the statistics literature in the context of "Fisher Scoring" (see Wang [2010] for a review).

[3]The equality can easily be verified by plugging the split (14) into the definition of the GGN (Eq. 10) and observing that $\nabla_b^2 a_n(b) = \nabla_b a_n(b) \, \nabla_b a_n(b)^\top$ as a special property of the choice $a_n(b) = -\log(b)$.

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
