[Supplementary Material · appendix.pdf]

# Limitations of the Empirical Fisher Approximation for Natural Gradient Descent
# Supplementary Material

## A  Details on natural gradient descent

We give an expanded version of the introduction to natural gradient descent provided in Section 3.1

### A.1  Measuring distance in Kullback-Leibler divergence

Gradient descent minimizes the objective function by updating in the "direction of steepest descent". But what, precisely, is meant by the direction of steepest descent? Consider the following definition,

$$\lim_{\varepsilon \to 0} \tfrac{1}{\varepsilon} \left( \arg\min_{\delta} f(\theta + \delta) \right) \quad \text{s.t.} \quad d(\theta, \theta + \delta) \le \varepsilon, \tag{20}$$

where $d(\cdot, \cdot)$ is some distance function. We are looking for the update step $\delta$ which minimizes $f$ within an $\varepsilon$ distance around $\theta$, and let the radius $\varepsilon$ go to zero (to make $\delta$ finite, we have to divide by $\varepsilon$). This definition makes clear that the direction of steepest descent is intrinsically tied to the geometry we impose on the parameter space by the definition of the distance function. If we choose the Euclidean distance $d(\theta, \theta') = \|\theta - \theta'\|_2$, Eq. (20) reduces to the (normalized) negative gradient.

Now, assume that $\theta$ parameterizes a statistical model $p_\theta(z)$. The parameter vector $\theta$ is not the main quantity of interest; the distance between $\theta$ and $\theta'$ would be better measured in terms of distance between the distributions $p_\theta$ and $p_{\theta'}$. A common function to measure the difference between probability distributions is the Kullback–Leibler (KL) divergence. If we choose $d(\theta, \theta') = D_{KL}\left(p_{\theta'} \,\|\, p_\theta\right)$, the steepest descent direction becomes the natural gradient, $\mathrm{F}(\theta)^{-1} \nabla \mathcal{L}(\theta)$, where

$$\mathrm{F}(\theta) = \nabla^2_{\theta'} \, D_{KL}\left(p_\theta \,\|\, p_{\theta'}\right)|_{\theta'=\theta}, \tag{21}$$

the Hessian of the KL divergence, is the *Fisher information matrix* of the statistical model and

$$\mathrm{F}(\theta) := \mathbb{E}_{p_\theta(z)}\left[\nabla \log p_\theta(z) \nabla \log p_\theta(z)^T\right] = \mathbb{E}_{p_\theta(z)}\left[-\nabla^2 \log p_\theta(z)\right] \tag{22}$$

To see why, apply the chain rule on the $\log$ to split the equation in terms of the Hessian and the outer product of the gradients of $p_\theta$ w.r.t. $\theta$,

$$\mathbb{E}_{p_\theta(z)}\left[-\nabla^2_\theta \log p_\theta(z)\right] = \mathbb{E}_{p_\theta(z)}\left[-\tfrac{1}{p_\theta(z)}\nabla^2_\theta p_\theta(z)\right] + \mathbb{E}_{p_\theta(z)}\left[\tfrac{1}{p_\theta(z)^2}\nabla_\theta p_\theta(z)\nabla_\theta p_\theta(z)^\top\right]. \tag{23}$$

The first term on the right-hand side is zero, since

$$\mathbb{E}_{p_\theta(z)}\left[-\tfrac{1}{p_\theta(z)}\nabla^2_\theta p_\theta(z)\right] := -\int_z \frac{1}{p_\theta(z)}\nabla^2_\theta p_\theta(z)p_\theta(z)\,\mathrm{d}z = \int_z \nabla^2_\theta p_\theta(z)\,\mathrm{d}z,$$

$$= \nabla^2_\theta \int_z p_\theta(z)\,\mathrm{d}z = \nabla^2_\theta[1] = 0. \tag{24}$$

The second term is the expected outer-product of the gradients, as $\partial_\theta \log f(\theta) = \tfrac{1}{f(\theta)}\partial_\theta f(\theta)$,

$$\tfrac{1}{p_\theta(z)^2}\nabla_\theta p_\theta(z)\nabla_\theta p_\theta(z)^\top = \left(\tfrac{1}{p_\theta(z)}\nabla_\theta p_\theta(z)\right)\left(\tfrac{1}{p_\theta(z)}\nabla_\theta p_\theta(z)\right)^\top,$$

$$= \nabla_\theta \log p_\theta(z)\,\nabla_\theta \log p_\theta(z)^\top. \tag{25}$$

The same technique also shows that if the empirical distribution over the data is equal to the model distribution $p_\theta(y|f(x, \theta))$, then the Fisher, empirical Fisher and Hessian are all equal.

## A.2 The Fisher for common loss functions

For a probabilistic conditional model of the form $p(y|f(x,\theta))$ where $p$ is an exponential family distribution, the equivalence between the Fisher and the generalized Gauss-Newton leads to a straightforward way to compute the Fisher without expectations, as

$$\mathrm{F}(\theta) = \sum_n (\mathrm{J}_\theta f(x_n,\theta))^\top (\nabla^2 \log p(y_n|f(x_n,\theta)))(\mathrm{J}_\theta f(x_n,\theta)) = \sum_n J_n^\top H_n J_n, \quad (26)$$

where $J_n = \mathrm{J}_\theta f(x_n,\theta)$ and $H_n = \nabla^2 \log p(y_n|f(x_n,\theta))$ often has an exploitable structure.

**The squared-loss** used in regression, $\frac{1}{2}\sum_n \|y_n - f(x_n,\theta)\|^2$, can be cast in a probabilistic setting with a Gaussian distribution with unit variance, $p(y_n|f(x_n,\theta)) = \mathcal{N}\left(y_n; f(x_n,\theta), 1\right)$,

$$p(y_n|f(x_n,\theta)) = \exp\left(-\tfrac{1}{2}\|y_n - f(x_n,\theta)\|^2\right). \quad (27)$$

The Hessian of the negative log-likelihood w.r.t. $f$ is then

$$\nabla_f^2 - \log p(y_n|f) = \nabla_f^2\left[-\log \exp\left(-\tfrac{1}{2}\|y_n - f\|^2\right)\right] = \nabla_f^2\left[\tfrac{1}{2}\|y_n - f\|^2\right] = 1. \quad (28)$$

And as the function $f$ is scalar-valued, the Fisher reduces to an outer-products of gradients,

$$\mathrm{F}(\theta) = \sum_n \nabla_\theta f(x_n,\theta)\nabla_\theta f(x_n,\theta)^\top. \quad (29)$$

We stress that this is difference to the outer product of gradients of the overall loss;

$$\mathrm{F}(\theta) \neq \sum_n \nabla_\theta \log p(y_n|f(x_n,\theta))\nabla_\theta \log p(y_n|f(x_n,\theta))^\top. \quad (30)$$

**The cross-entropy loss** used in $C$-class classification can be cast as an exponential family distribution by using the softmax function on the mapping $f(x_n,\theta)$,

$$p(y_n = c|f(x_n,\theta)) = [\mathrm{softmax}(f)]_c = \frac{e^{f_c}}{\sum_i e^{f_i}} = \pi_c, \quad (31)$$

The Hessian of the negative log-likelihood w.r.t. $f$ is independent of the class label $c$,

$$\nabla_f^2(-\log p(y = c|f)) = \nabla_f^2[-f_c + \log\left(\sum_i e^{f_i}\right)] = \nabla_f^2[\log\left(\sum_i e^{f_i}\right)]. \quad (32)$$

A close look at the partial derivatives shows that

$$\frac{\partial^2}{\partial f_i^2}\log\left(\sum_c e^{f_c}\right) = \frac{e^{f_i}}{(\sum_c e^{f_c})} - \frac{e^{f_i}{}^2}{(\sum_c e^{f_c})^2}, \quad \text{and} \quad \frac{\partial^2}{\partial f_i \partial f_j}\log\left(\sum_c e^{f_c}\right) = -\frac{e^{f_i}e^{f_j}}{(\sum_c e^{f_c})^2}, \quad (33)$$

and the Hessian w.r.t. $f$ can be written in terms of the vector of predicted probabilities $\pi$ as

$$\nabla_f^2(-\log p(y|f)) = \mathrm{diag}(\pi) - \pi\pi^\top. \quad (34)$$

Writing $\pi_n$ the vector of probabilities associated with the $n$th sample, the Fisher becomes

$$\mathrm{F}(\theta) = \sum_n [\mathrm{J}_\theta f(x_n,\theta)]^\top(\mathrm{diag}(\pi_n) - \pi_n\pi_n^\top)[\mathrm{J}_\theta f(x_n,\theta)]. \quad (35)$$

## A.3 The generalized Gauss-Newton as a linear approximation of the model

In Section 3.3, we mentioned that the generalized Gauss-Newton with a split $\mathcal{L}(\theta) = \sum_n a_n(b_n(\theta))$ can be interpreted as an approximation of $\mathcal{L}$ where the second-order information of $a_n$ is conserved but the second-order information of $b_n$ is ignored. To make this connection explicit, see that if $b_n$ is a linear function, the Hessian and the GGN are equal as the Hessian of $b_n$ w.r.t. to $\theta$ is zero,

$$\nabla^2 \mathcal{L}(\theta) = \underbrace{\sum_n (\mathrm{J}_\theta b_n(\theta))^\top \; \nabla_b^2 a_n(b_n(\theta)) \; (\mathrm{J}_\theta b_n(\theta))}_{\text{GGN}} + \sum_{n,m}[\nabla_b a_n(b_n(\theta))]_m \underbrace{\nabla_\theta^2 b_n^{(m)}(\theta)}_{=0}. \quad (36)$$

This corresponds to the Hessian of a local approximation of $\mathcal{L}$ where the inner function $b$ is linearized. We write the first-order Taylor approximation of $b_n$ *around* $\theta$ as a function of $\theta'$,

$$\bar{b}_n(\theta,\theta') := b_n(\theta) + \mathrm{J}_\theta b_n(\theta)(\theta' - \theta),$$

and approximate $\mathcal{L}(\theta')$ by replacing $b_n(\theta')$ by its linear approximation $\bar{b}_n(\theta,\theta')$. The generalized Gauss-Newton is the Hessian of this approximation, evaluated at $\theta' = \theta$,

$$G(\theta) = \nabla_{\theta'}^2 \sum_n a_n(\bar{b}_n(\theta,\theta'))|_{\theta'=\theta} = \sum_n (\mathrm{J}_\theta b_n(\theta))^\top \nabla_b^2 a_n(b_n(\theta)) \; (\mathrm{J}_\theta b_n(\theta)) \quad (37)$$

## B  Computational aspects

The empirical Fisher approximation is often motivated as an easier-to-compute alternative to the Fisher. While there is some merit to this argument, we argued in the main text that it computes the wrong quantity. A Monte Carlo approximation to the Fisher has the same computational complexity and a similar implementation: sample one output $\tilde{y}_n$ from the model distribution $p(y|f(x_n, \theta))$ for each input $x_n$ and compute the outer product of the gradients

$$\sum_n \nabla \log p(\tilde{y}_n|f(x_n, \theta)) \nabla \log p(\tilde{y}_n|f(x_n, \theta))^\top. \tag{38}$$

While noisy, this one-sample estimate is unbiased and does not suffer from the problems mentioned in the main text. This is the approach used by Martens and Grosse [2015] and Zhang et al. [2018].

As a side note, some implementations use a biased estimate by using the most likely output $\hat{y}_n = \arg\max_y p(y|f(x_n, \theta))$ instead of sampling $\tilde{y}_n$ from $p(y|f(x_n, \theta))$. This scheme could be beneficial in some circumstances as it reduces variance, but it can backfire by increasing the bias. For the least-squares loss, $p(y|f(x_n, \theta))$ is a Gaussian distribution centered as $f(x_n, \theta)$ and the most likely output is $f(x_n, \theta)$. The gradient $\nabla_\theta \log p(y|f(x_n, \theta))|_{y=f(x_n,\theta)}$ is then always zero.

For high quality estimates, sampling additional outputs and averaging the results is inefficient. If $M$ MC samples $\tilde{y}_1, \ldots, \tilde{y}_M$ per input $x_n$ are used to compute the gradients $g_m = \nabla \log p(\tilde{y}_m|f(x_n, \theta))$, most of the computation is repeated. The gradient $g_m$ is

$$g_m = \nabla \log p(\tilde{y}_m|f(x_n, \theta)) = -(\mathrm{J}_\theta f(x_n, \theta))^\top \nabla_f \log p(\tilde{y}_m|f), \tag{39}$$

where the Jacobian of the model output, $\mathrm{J}_\theta f$, does not depend on $\tilde{y}_m$. The Jacobian of the model is typically more expensive to compute than the gradient of the log-likelihood w.r.t. the model output, especially when the model is a neural network. This approach repeats the difficult part of the computation $M$ times. The expectation can instead be computed in closed form using the generalized Gauss-Newton equation (Eq. 26, or Eq. 10 in the main text), which requires the computation of the Jacobian only once per sample $x_n$.

The main issue with this approach is that computing Jacobians is currently not well supported by deep learning auto-differentiation libraries, such as TensorFlow or Pytorch. However, the current the implementations relying on the empirical Fisher also suffer from this lack of support, as they need access to the individual gradients to compute their outer-product. Access to the individual gradients is equivalent to computing the Jacobian of the vector $[-\log p(y_1|f(x_1, \theta)), ..., -\log p(y_N|f(x_N, \theta)]^\top$. The ability to efficiently compute Jacobians and/or individual gradients in parallel would drastically improve the practical performance of methods based on the Fisher and empirical Fisher, as most of the computation of the backward pass can be shared between samples.

## C  Additional proofs

### C.1  Proof of Propositon 1

In Section 3.4, Proposition 1, we stated that the Fisher and the generalized Gauss-Newton are equivalent for the problems considered in the introduction;

> **Proposition 1** (Martens [2014], §9.2). *If $p(y|f)$ is an exponential family distribution with natural parameters $f$, then the Fisher information matrix coincides with the GGN of Eq.* (1) *using the split*
> $$a_n(b) = -\log p(y_n|b), \qquad\qquad b_n(\theta) = f(x_n, \theta),$$
> *and reads* $\mathrm{F}(\theta) = G(\theta) = \sum_n [\mathrm{J}_\theta f(x_n, \theta)]^\top \nabla_f^2 \log p(y_n|f(x_n, \theta)) [\mathrm{J}_\theta f(x_n, \theta)].$

Plugging the split into the definition of the GGN (Eq. 10) yields $G(\theta)$, so we only need to show that the Fisher coincides with this GGN. By the chain rule, we have

$$\nabla_\theta \log p(y|f(x_n, \theta)) = \mathrm{J}_\theta f(x_n, \theta)^\top \nabla_f \log p(y|f(x_n, \theta)), \tag{40}$$

and we can then apply the following steps.

$$\mathrm{F}(\theta) = \sum_n \mathbb{E}_{y \sim p_\theta(y|x_n)} \left[ \mathrm{J}_\theta f(x_n, \theta)^\top \nabla_f \log p(y|f_n) \nabla_f \log p(y|f_n)^\top \mathrm{J}_\theta f(x_n, \theta) \right], \tag{41}$$

$$= \sum_n \mathrm{J}_\theta f(x_n, \theta)^\top \mathbb{E}_{y \sim p_\theta(y|x_n)} \left[ \nabla_f \log p(y|f_n) \nabla_f \log p(y|f_n)^\top \right] \mathrm{J}_\theta f(x_n, \theta), \tag{42}$$

$$= \sum_n \mathrm{J}_\theta f(x_n, \theta)^\top \mathbb{E}_{y \sim p_\theta(y|x_n)} \left[ -\nabla_f^2 \log p(y|f_n) \right] \mathrm{J}_\theta f(x_n, \theta), \tag{43}$$

Eq. (41) rewrites the Fisher using the chain rule, Eq. (42) take the Jacobians out of the expectation as they do not depend on $y$ and Eq. (43) is due to the equivalence between the expected outer product of gradients and expected Hessian shown in the last section.

If $p$ is an exponential family distribution with natural parameters (a linear combination of) $f$, its log density has the form $\log p(y|f) = f^T T(y) - A(f) + \log h(y)$ where $T$ are the sufficient statistics, $A$ is the cumulant function, and $h$ is the base measure. Its Hessian w.r.t. $f$ is independent of $y$,

$$\mathrm{F}(\theta) = \sum_n \mathrm{J}_\theta f(x_n, \theta)^\top \nabla_f^2 (-\log p(y_n|f_n)) \mathrm{J}_\theta f(x_n, \theta), \tag{44}$$

## C.2   Proof of Proposition 2

In §3.4, Prop. 2, we show that the difference between the Fisher (or the GNN) and the Hessian can be bounded by the residuals and the smoothness constant of the model $f$;

> **Proposition 2.** *Let $\mathcal{L}(\theta)$ be defined as in Eq. (1) with $\mathbb{F} = \mathbb{R}^M$. Denote by $f_n^{(m)}$ the $m$-th component function of $f(x_n, \cdot)\colon \mathbb{R}^D \to \mathbb{R}^M$ and assume each $f_n^{(m)}$ is $\beta$-smooth. Let $G(\theta)$ be the GGN (Eq. 10). Then,*
>
> $$\left\|\nabla^2 \mathcal{L}(\theta) - G(\theta)\right\|_2^2 \leq r(\theta)\beta, \tag{45}$$
>
> *where $r(\theta) = \sum_{n=1}^{N} \|\nabla_f \log p(y_n|f(x_n, \theta))\|_1$ and $\|\cdot\|_2$ denotes the spectral norm.*

Dropping $\theta$ from the notation for brevity, the Hessian can be expressed as

$$\nabla^2 \mathcal{L} = G + \sum_{n=1}^{N} \sum_{m=1}^{M} r_n^{(m)} \nabla_\theta^2 f_n^{(m)}, \qquad \text{where} \qquad r_n^{(m)} = \frac{\partial \log p(y_n|f)}{\partial f^{(m)}}\Big|_{f=f_n(\theta)} \tag{46}$$

is the derivative of $-\log p(y|f)$ w.r.t. the $m$-th component of $f$, evaluated at $f = f_n(\theta)$.

If all $f_n^{(m)}$ are $\beta$-smooth, their Hessians are bounded by $-\beta I \preceq \nabla_\theta^2 f_n^{(m)} \preceq \beta I$ and

$$-\left|\sum_{n,m} r_n^{(m)}\right| \beta\, \mathbf{I} \preceq \nabla^2 \mathcal{L} - G \preceq \left|\sum_{n,m} r_n^{(m)}\right| \beta\, \mathbf{I}. \tag{47}$$

Pulling the absolute value inside the double sum gives the upper bound

$$\left|\sum_{n,m} r_n^{(m)}\right| \leq \sum_n \sum_m \left|\frac{\partial \log p(y_n|f)}{\partial f^{(m)}}\Big|_{f=f_n(\theta)}\right| = \sum_n \|\nabla_f \log p(y_n|f_n(\theta))\|_1, \tag{48}$$

and the statement about the spectral norm (the largest singular value of the matrix) follows.

# D   Experimental details

In contrast to the main text of the paper, which uses the sum formulation of the loss function,

$$\mathcal{L}(\theta) = \sum_n \log p(y_n|f(x_n, \theta)),$$

the implementation—and thus the reported step sizes and damping parameters—apply to the average,

$$\mathcal{L}(\theta) = \tfrac{1}{N} \sum_n \log p(y_n|f(x_n, \theta)).$$

The Fisher and empirical Fisher are accordingly rescaled by a $1/N$ factor.

## D.1   Vector field of the empirical Fisher preconditioning

The problem used for Fig. 1 is a linear regression on $N = 1000$ samples from

$$x_i \sim \text{Lognormal}\left(0, 3/4\right), \qquad \epsilon_i \sim \mathcal{N}\left(0, 1\right), \qquad y_i = 2 + 2x_i + \epsilon_i. \tag{49}$$

To be visible and of a similar scale, the gradient, natural gradient and empirical Fisher-preconditioned gradient were relatively rescaled by $1/3$, $1$ and $3$, respectively. The trajectories of each method is computed by running each update,

$$\text{GD:} \qquad \theta_{t+1} = \theta_t - \gamma \nabla \mathcal{L}(\theta_t). \tag{50}$$

$$\text{NGD:} \qquad \theta_{t+1} = \theta_t - \gamma (\mathrm{F}(\theta_t) + \lambda\, \mathbf{I})^{-1} \nabla \mathcal{L}(\theta_t), \tag{51}$$

$$\text{EFGD:} \qquad \theta_{t+1} = \theta_t - \gamma (\widetilde{\mathrm{F}}(\theta_t) + \lambda\, \mathbf{I})^{-1} \nabla \mathcal{L}(\theta_t), \tag{52}$$

using a step size of $\gamma = 10^{-4}$ and a damping parameter of $\lambda = 10^{-8}$ to ensure stability for $50'000$ iterations. The vector field is computed using the same damping parameter. The starting points are

$$\begin{bmatrix} 2 & 4.5 \end{bmatrix}, \qquad \begin{bmatrix} 1 & 0 \end{bmatrix}, \qquad \begin{bmatrix} 4.5 & 3 \end{bmatrix}, \qquad \begin{bmatrix} -0.5 & 3 \end{bmatrix}.$$

## D.2 EF as a quadratic approximation at the minimum for misspecified models

The problems are optimized using using the Scipy [Jones et al., 2001] implementation of BFGS[4]. The quadratic approximation of the loss function using the matrix $M$ (the Fisher or empirical Fisher) used is $\mathcal{L}(\theta) \approx \frac{1}{2}(\theta - \theta^\star)M(\theta - \theta^\star)$, for $\|\theta - \theta^\star\|^2 = 1$. The datasets used for the logistic regression problem of Fig. 2 are described in Table 2. Fig. 4 shows additional examples of model misspecification on a linear regression problem using the datasets described in Table 3. All experiments used $N = 1'000$ samples.

Table 2: Datasets used for Fig. 2. For all datasets, $p(y = 0) = p(y = 1) = 1/2$.

| Model | $p(x\|y=0)$ | $p(x\|y=1)$ |
|---|---|---|
| Correct model: | $\mathcal{N}\left(\begin{bmatrix}1\\1\end{bmatrix}, \begin{bmatrix}2&0\\0&2\end{bmatrix}\right)$ | $\mathcal{N}\left(\begin{bmatrix}-1\\-1\end{bmatrix}, \begin{bmatrix}2&0\\0&2\end{bmatrix}\right)$ |
| Misspecified (A): | $\mathcal{N}\left(\begin{bmatrix}1.5\\1.5\end{bmatrix}, \begin{bmatrix}3&0\\0&3\end{bmatrix}\right)$ | $\mathcal{N}\left(\begin{bmatrix}-1.5\\-1.5\end{bmatrix}, \begin{bmatrix}1&0\\0&1\end{bmatrix}\right)$ |
| Misspecified (B): | $\mathcal{N}\left(\begin{bmatrix}-1\\-1\end{bmatrix}, \begin{bmatrix}1.5&-0.9\\-0.9&1.5\end{bmatrix}\right)$ | $\mathcal{N}\left(\begin{bmatrix}1\\1\end{bmatrix}, \begin{bmatrix}1.5&0.9\\0.9&1.5\end{bmatrix}\right)$ |

Table 3: Datasets used for Fig. 4. For all datasets, $x \sim \mathcal{N}(0, 1)$.

| Model | $y$ | $\epsilon$ |
|---|---|---|
| Correct model: | $y = x + \epsilon$ | $\epsilon \sim \mathcal{N}(0, 1)$ |
| Misspecified (A): | $y = x + \epsilon$ | $\epsilon \sim \mathcal{N}(0, 2)$ |
| Misspecified (B): | $y = x + \frac{1}{2}x^2 + \epsilon$ | $\epsilon \sim \mathcal{N}(0, 1)$ |

## D.3 Optimization with the empirical Fisher as preconditioner

The optimization experiment uses the update rules described in §D.1 by Eq. (50, 51, 52). The step size and damping hyperparameters are selected by a gridsearch, selecting for each optimizer the run with the minimal loss after 100 iterations. The grid used is described in Table 5 as a log-space[5]. Table 4 describes the datasets used and Table 6 the hyperparameters selected by the gridsearch. The cosine similarity is computed between the gradient preconditioned with the empirical Fisher and the Fisher, without damping, at each step along the path taken by the empirical Fisher optimizer.

The problems are initialized at $\theta_0 = 0$ and run for 100 iterations. This initialization is favorable to the empirical Fisher for the logistic regression problems. Not only is it guaranteed to not be arbitrarily wrong, but the empirical Fisher and the Fisher coincide when the predicted probabilities are uniform. For the sigmoid activation of the output of the linear mapping, $\sigma(f)$, the gradient and Hessian are

$$-\frac{\partial}{\partial f}\log p(y|f) = \sigma(f) \qquad\qquad -\frac{\partial^2}{\partial f^2}\log p(y|f) = \sigma(f)(1 - \sigma(f)). \qquad (53)$$

They coincide when $\sigma(f) = \frac{1}{2}$, at $\theta = 0$, or when $\sigma(f) \in \{0, 1\}$, which require infinite weights.

Table 4: Datasets

| Dataset | # Features | # Samples | Type | Figure |
|---|---|---|---|---|
| a1a[6] | $1'605$ | 123 | Classification | Fig. 3 |
| BreastCancer[7] | 683 | 10 | Classification | Fig. 3 |
| Boston Housing[8] | 506 | 13 | Regression | Fig. 3 |
| Yacht Hydrodynamics[9] | 308 | 7 | Regression | Fig. 5 |
| Powerplant[10] | $9'568$ | 4 | Regression | Fig. 5 |
| Wine[11] | 178 | 13 | Regression | Fig. 5 |
| Energy[12] | 768 | 8 | Regression | Fig. 5 |

Table 5: Grid used for the hyperparameter search for the optimization experiments, in $\log_{10}$. The number of samples to generate was selected as to generate a smooth grid in base 10, e.g., $10^0, 10^{.25}, 10^{.5}, 10^{.75}, 10^1, 10^{1.25}, \ldots$

| Parameter | | Grid |
|---|---|---|
| Step size | $\gamma$ | `logspace(start=-20, stop=10, num=241)` |
| Damping | $\lambda$ | `logspace(start=-10, stop=10, num=41)` |

Table 6: Selected hyperparameters, given in $\log_{10}$.

| Dataset | Algorithm | $\gamma$ | $\lambda$ | Dataset | Algorithm | $\gamma$ | $\lambda$ |
|---|---|---|---|---|---|---|---|
| Boston | GD | $-5.250$ | | Wine | GD | $-5.625$ | |
| | NGD | $0.125$ | $-10.0$ | | NGD | $0.000$ | $-8.5$ |
| | EFGD | $-1.250$ | $-8.0$ | | EFGD | $-1.375$ | $-6.0$ |
| BreastCancer | GD | $-5.125$ | | Energy | GD | $-5.500$ | |
| | NGD | $0.125$ | $-10.0$ | | NGD | $0.000$ | $-7.5$ |
| | EFGD | $-1.250$ | $-10.0$ | | EFGD | $0.875$ | $-3.0$ |
| a1a | GD | $0.250$ | | Powerplant | GD | $-5.750$ | |
| | NGD | $0.250$ | $-10.0$ | | NGD | $-0.625$ | $-8.0$ |
| | EFGD | $-0.375$ | $-8.0$ | | EFGD | $3.375$ | $-1.0$ |
| | | | | Yacht | GD | $-1.500$ | |
| | | | | | NGD | $-0.750$ | $-7.5$ |
| | | | | | EFGD | $1.625$ | $-6.5$ |

# E   Additional plots

Fig. 4 repeats the experiment described in Fig. 2 (§4.2), on the effect of model misspecification on the Fisher and empirical Fisher at the minimum, on linear regression problems instead of a classification problem. Similar issues in scaling and directions can be observed.

Fig. 5 repeats the experiment described in Fig. 3 (§4.3) on additional linear regression problems. Those additional examples show that the poor performance of empirical Fisher-preconditioned updates compared to NGD is not isolated to the examples shown in the main text.

Fig. 6 show the linear regression problem on the Boston dataset, originally shown in Fig. 3, where each line is a different starting point, using the same hyperparameters as in Fig. 3. The starting points are selected from $[-\theta^\star, \theta^\star]$, where $\theta^\star$ is the optimum. When the optimization starts close to the minimum (low loss), the empirical Fisher is a good approximation to the Fisher and there are very few differences with NGD. However, when the optimization starts far from the minimum (high loss), the individual gradients, and thus the sum of outer product gradients, are large, which leads to very small steps, regardless of curvature, and slow convergence. While this could be counteracted with a larger step size in the beginning, this large step size would not work close to the minimum and would lead to oscillations. The selection of the step size therefore depends on the starting point, and would ideally be on a decreasing schedule.

Figure 4: Quadratic approximations of the loss function using the Fisher and the empirical Fisher on a linear regression problem. The EF is a good approximation of the Fisher at the minimum if the data is generated by $y \sim \mathcal{N}(x\theta^* + b^*, 1)$, as the model assumes (left panel), but can be arbitrarily wrong if the assumption is violated, even at the minimum and with large N. In (A), the model is misspecified as it under-estimates the observation noise (data is generated by $y \sim \mathcal{N}(x\theta^* + b^*, 2)$). In (B), the model is misspecified as it fails to capture the quadratic relationship between $x$ and $y$.

Figure 5: Comparison of the Fisher (NGD) and the empirical Fisher (EFGD) as preconditioners on additional linear regression problems. The second row shows the cosine similarity between the EF-preconditioned gradient and the natural gradient at each step on the path taken by EFGD.

Figure 6: Linear regression on the Boston dataset with different starting points (each line is a different initialization). When the optimization starts close to the minimum (low initial loss), the empirical Fisher is a good approximation to the Fisher and there are very few differences with NGD, but the performance degrades as the optimization procedure starts farther away (large initial loss).

## Footnotes

[4] https://docs.scipy.org/doc/scipy/reference/optimize.minimize-bfgs.html

[5] https://docs.scipy.org/doc/numpy/reference/generated/numpy.logspace.html

[6] `www.csie.ntu.edu.tw/ cjlin/libsvmtools/datasets/binary.html#a1a`

[7] `www.csie.ntu.edu.tw/c̃jlin/libsvmtools/datasets/binary.html#breast-cancer`

[8] `scikit-learn.org/stable/modules/generated/sklearn.datasets.load_boston.html`

[9] `archive.ics.uci.edu/ml/datasets/Yacht+Hydrodynamics`

[10] `archive.ics.uci.edu/ml/datasets/Combined+Cycle+Power+Plant`

[11] `archive.ics.uci.edu/ml/datasets/Wine`

[12] `archive.ics.uci.edu/ml/datasets/Energy+efficiency`