[Reviews · NeurIPS 2019]

Reviewer 1



Originality: While the paper does not propose a novel algorithm, it presents an in-depth discussion of the (lack of a) relationship between the EF and F despite the seeming similarity. From a scientific standpoint I find this kind of contribution that strengthens the understanding of a family of methods and raises questions for new avenues of research substantially more valuable than yet another "state-of-the-art" algorithm on some arbitrary benchmark that provides little insight into why it works. Quality: The write-up is clearly focussed with a thoughtfully chosen set of empirical examples and experiments. I think this is a well-executed paper that clearly warrants publication in NeurIPS. Clarity: The paper is well written and structured, and overall easy to follow. I appreciate that crucial ideas are re-emphasized throughout the paper, making it easy to keep them in mind throughout and when relevant, and that existing work is credited clearly. Significance: The paper raises some overlooked subtle, but, as demonstrated, important issues. I therefore believe that this is a significant piece of work, even though second-order optimization is perhaps somewhat niche at the moment.

Reviewer 2



Originality: the paper lacks a sound and novel contribution. Theoretically, there is only one minor result as stated above. Technically, there is not a systematical experimental study on real deep networks. The main contribution is on discussing two different formulations of the Fisher matrix. The main trick on making these two formulations different (despite that the authors took a sophisticated approach going though GGN) is that the so called empirical Fisher relies on y_n (target of neural network output), and if one consider y_n to be randomly distributed with fixed variance based on the neural network output, the two formulations are equivalent, otherwise there is a scale parameter in eq.(3) which is shrinking making the two formulations different because of the shrinking and damping. In practice, for applying the exact natural gradient (without approximation) to deep neural networks, there is no reason to use the EF over the Fisher. For example, in the cited "revisiting natural gradient" paper, the Fisher matrix is given in eq.(26), (28), etc. and does not depend on the target y_n, and one has no reason to use the EF formulation. Quality: the experiments are mainly performed on toy examples. It is not clear how these results can be generalized to real networks. The literature review can be enhanced, the authors used the term "empirical Fisher" which is used by the some machine learning papers (or is that the case the authors coined the term? please make clear on its first appearance) This should be connected to observed Fisher information which is a well-defined concept in statistics. There should be more discussions on information geometry and deep learning, where many literature are skipped. Clarity: the paper is well written both in English and in math. Significance: the paper can potentially broaden the audience of natural gradient methods to the deep learning community. However it has limited significance due to the lack of novel contributions. Overall I feel that it is below the bar of standard NeurIPS papers.

Reviewer 3



Originality: Such a paper is much needed as many authors blindly use the empirical FIM in place of other matrices in many applications Quality: section 3.1 would require some clarification. Clarity: the paper is well written and concise Significance: as already said, such a work that critically analyses the empirical FIM is much needed

Reviewer 4



Strengths --------- In terms of new material, - one contribution of the paper is to propose a refined definition of a generalized Gauss-Newton matrix. This could open new convergence analyzes but it is not pursued in the present paper. - numerous experiments are presented to compare the empirical fisher matrix to the true one in terms of preconditioners for optimization on simple models. These experiments support clearly the use of the true Fisher matrix. The code seems very well written and reusable. The real strength of the paper is pedagogical: - it is very well-written and illustrated (see Fig. 2 or 4 for example) - it has even more value that a lot of previous authors did not make any difference and made numerous false statements. Weaknesses ---------- 1. Except the new definition of the Generalized Gauss-Newton matrix (that is not pursued), no other proposition in the paper is original. 2. As the authors point themselves, analyzing the EF as a variance adaptation method would have explained its efficiency and strengthened the paper: "This perspective on the empirical Fisher is currently not well studied. Of course, there are obvious difficulties ahead:" Overcoming these difficulties is what a research paper is about, not only discussing them. 3. The main point of the paper relies in paragraph 3.2. This requires clear and sound propositions such as: for a well-specified model, and a consistent estimator, the empirical fisher matrix converges to the Hessian at a rate ... It is claimed to be specified in Appendix C.3 but there seems to be a referencing problem in the paper. This would highlight both the reasoning of previous papers and the difference with the actual approximation made here. Minor comments: --------------- Typos: - Eq. 5 no square for gradient of a _ n - Eq. 8 subscript theta should be under p not log - Replace the occurrences of Appendix A to Appendix C Conclusion: ---------- Overall I think this is an good lecture on natural gradient and its subtleties, yet not a research paper since almost no new results are demonstrated. Yet, if the choice has to be made between another paper that uses the empirical Fisher and this one that explains it, I'll advocate for this paper. Therefore I tend to marginally accept this paper though I think its place is in lecture notes (in fact Martens long review of natural gradient [New insights and perspectives on the natural gradient method, Martens 2014] should incorporate it, that is where this paper should be from my opinion.) After discussion -------------------- After the discussion, I increased my score, I don't think that it is a top paper as it does not have new results but it should clearly be accepted as it would be much more helpful than "another state of the art technique for deep learning" with some misleading approximations like ADAM. Note that though refining the definition of a generalized gauss-newton method seems to be a detail, I think it could have a real potential for further analysis in optimization.

[Author Response · NeurIPS 2019]

**We thank all reviewers for their time and expertise. We are particularly happy that R1 and R3 are positive**
**about our paper. R1's summary of our contributions resonates strongly with what we wanted to achieve.**

**We are worried that R2 may have misunderstood main points of the paper. Some of their comments seem to**
**suggest that we endorse the EF. The exact opposite is the case. The EF, which doesn't exist in the Stats literature,**
**is widely used in ML (see our citations). We argue that it should *not* be used. Let us clarify some of your points:**

> (R2) *"for applying the exact natural gradient [...] there is no reason to use the EF over the Fisher."*
We agree! However, a large portion of the literature applying natural gradient ideas to machine learning uses the EF and
many researchers are under the impression that the EF and the Fisher are interchangeable. We cite 15 papers that use
the EF, 9 of them from the past 3 years, and we know of at least 4 additional papers, accepted or under review at ICML,
UAI and NeurIPS 2019, perpetuating this confusion. Basically, our paper makes the points that you levy against us in
the review. Our goal is to point out the differences between the EF and the Fisher and clear the confusion, in the hope
that future applications of natural gradient methods to machine learning will be more impactful.

> (R2) *"the authors used the term "empirical Fisher" [... did the authors coin] the term?"*
We did not coin this term and, as noted in L265-267, we believe it is a misnomer. To the best of our knowledge, the
term "empirical Fisher" first appears in the manuscript of Martens [2014]. Altough the origin of the concept might be
Le Roux et al. [2007], who confused the covariance and the Fisher information, and Graves [2011] who proposed it as
an approximation of the Hessian. The idea was heavily popularized by the Adam optimizer [Kingma and Ba, 2015].

> (R2) *"This should be connected to observed Fisher information which is a well-defined concept in statistics."*
This is a fantastic point that we definitely need to address. While this terminology is widely used in the literature we
cite, it is true that the concept might sound strange to statisticians as the terms "observed Fisher" and, even more
confusing, "empirical Fisher information", are used, in statistics, to refer to what the community we are trying to reach
calls the Fisher information. We will use the additional page to make the context clearer to a larger audience and give
an overview of how the EF approximation came to be.

> (R2) *"the experiments are mainly performed on toy examples. It is not clear how [to generalize] to real networks"*
Note again that we are arguing *against* the use of the EF concept. If it doesn't even work on toy problems, why rely on
it in "real networks" that are much harder to understand?

**Questions raised by R3 & R1 (thanks for these!):**
> (R3) *"the plots in fig 3 show that the cosine [...] is close to 1 near the end of the training, which seem to contradict*
*the claims in subsection 3.1 and 3.2: in practice it seems that the cases discussed in 3.1 and 3.2 are not met."*
The plots for §3.1 and 3.2 are in Fig.2, showing strong differences at the minimum. Fig. 3 showcases §3.3, about
preconditioning, showing that updates can be opposite. While the cosine gets close to 1 in Fig.3, this metric captures
only a small part of the relationship and the two matrices can still be very different (see figure at end of this rebuttal).

> (R3) *"how do you obtain eq. 12? [...] line 184 the Fisher does not match eq. 2"* Eq.12 and L184 show the equations
for the empirical Fisher (Eq.3) and the Fisher (Eq.2) applied to the linear regression example of L178-179.

> (R3) *"line 192 if b is arbitrarily close to 1 then $\nabla_b \log b$ is arbitrarily close to 0 [...] I think you wanted to write that*
*using the definition line 179 with a variance 1 normal distribution, $b_n(\theta)$ can not get arbitrarily close to 1."* and
> (R1) *"For classification [when] all labels are predicted correctly with probability 1, wouldn't [F = EF = 0?]"*
The subtelty of the proposed change to the definition of the GGN (Prop.3) relies on this key point: for the guarantee in
Prop.2 to hold, $\nabla_b a(b)$ needs to go to 0, which can not happen for $a(b) = \log b$ as $\nabla_b \log b = 1/b$ (for $b \in [0,1]$). As
both reviewers noted, if $b$ goes to 1 and is correct, the EF is 0 (the gradients are 0), the Fisher is 0 (the model is no
longer probabilistic, meaning new samples can not give more information) and the Hessian is also 0 (some parameters,
the weights for classification or the precision–if learned–for regression, need to reach $\infty$ and the landscape is infinitely
flat). We will add a paragraph to explain this special case.

> (R3) *"in eq 15 I think the multiplicative N should be in front of the covariance matrix if we follow [...] eq 3"*
We defined $\Sigma$ as the covariance a uniformly distributed gradient scaled by $N$ (L184) to be consistent with the definition
of the loss as a sum in Eq.1. Sorry for the confusion; we will try to make it clearer.

**Typos found by R3: Will all be addressed, sincere thanks for the close reading.** (Eq.5, L.183-185 and Prop.2)
Sorry about the lost pointer to the proof of Prop.2 - it is in the supplementary §C.3.

Quadratic approximations (as in Fig.2) using the projection of F (yellow) and EF (red) on the two largest eigenvectors of F, at the end of training with the EF, using the settings of Fig.3. While the cosines are close to 1, the matrices can still be very different in terms of directions and scaling.

[Meta-Review · NeurIPS 2019]

All reviewers were positive about the paper. The paper corrects several common incorrect assertions and misleading derivations in the natural gradient algorithms literature. The exposition is remarkably clear, with a potential to serve as a reference paper on the topic. The paper is clearly of broad interest to the machine learning community. We recommend to take the reviewers' comments and suggestions into account while preparing the camera ready final version of the paper. The authors might also want to consider an alternative title that would better reflect the scope of the paper and in particular better support its potential to be a reference paper for researchers interested in natural gradient algorithms. Accept.